# A Scoping Review of the Validity, Reliability and Conceptual Alignment of Food Literacy Measures for Adults

**DOI:** 10.3390/nu11040801

**Published:** 2019-04-08

**Authors:** Claudia Amouzandeh, Donna Fingland, Helen Anna Vidgen

**Affiliations:** School of Exercise and Nutrition Sciences, Queensland University of Technology, Victoria Park Road, Kelvin Grove, QLD 4059, Australia; claudia@qccs.com.au (C.A.); donnafingland@icloud.com (D.F.)

**Keywords:** food literacy, measurement, assessment tool, questionnaire, validity, reliability, evaluation

## Abstract

The measurement of food literacy has recently gained momentum globally. The aim of this paper is to review the literature in order to describe and analyse the measurement of adult food literacy. The objectives are to i) identify tools that explicitly measure food literacy in adults; ii) summarise their psychometric properties; and iii) critique tool items against the four domains and 11 components of food literacy, as conceptualised by Vidgen and Gallegos. Using the PRISMA guidelines, a search of seven databases (PubMed, Embase, ScienceDirect, Scopus, EBSCOhost, A+ Education, and ProQuest) was undertaken. 12 studies met the inclusion criteria. Papers reported on either the development of a tool to explicitly measure food literacy or a part thereof (*n* = 5); food literacy strategy indicators (*n* = 1); tools developed to evaluate a food literacy intervention (*n* = 3); or tools to measure food literacy as a characteristic within a broader study (*n* = 3). Six tools captured all four domains. None measured all components. Items measuring the same component varied considerably. Most tools referenced a theoretical framework, were validated and reliable. This review will assist practitioners select and develop tools for the measurement of food literacy in their context.

## 1. Introduction

Increasing rates of diet-related disease has been linked to an apparent decline in the general population’s food knowledge and skills [1,2]. A plethora of commentaries on this association exist in the literature, with authors describing a “gastronomic revolution” [3], an “epidemic of culinary ineptness” [4] and a “dietary cacophony" of conflicting information that “deadens” an individual’s capacity to eat [5,6]. Indeed, as society’s foodscapes become increasingly complex, there is concern that we are individually and collectively becoming increasingly “de-skilled” and no longer possess fundamental food skills and practices for healthy eating [1,7,8,9]. 

In light of these issues, the concept of ‘food literacy’ has emerged as an integrative framework and approach to describe the relevant knowledge, skills and behaviours necessary to achieve a diet aligned with nutrition recommendations. In 2014, Vidgen and Gallegos empirically defined food literacy as “a collection of inter-related knowledge, skills and behaviours required to plan, manage, select, prepare and eat foods to meet needs and determine food intake”, as well as, “the scaffolding that empowers individuals, households, communities or nations to protect diet quality through change and support dietary resilience over time” [2]. This definition significantly advanced the concept of food literacy and is widely cited as one of the most comprehensive food literacy definitions [10,11].

Along with increasing interest and clarity of food literacy, there has been a growing demand for comprehensive measurement tools [12]. The measurement of food literacy is important to test the conceptualisation of this new construct and its relationship to food intake [13]. Moreover, validated tools are required to monitor the food literacy of individuals and populations, plan and evaluate the effectiveness of interventions, and inform public health policy and practice [10,11,14]. Nevertheless, there is currently limited evidence on how food literacy is measured.

To date, very few studies have reviewed the development and application of existing food literacy measurement tools. A recent review by Yuen, Thomson & Gardiner [15] critically appraised the psychometric properties of 13 existing measures of adult nutrition literacy and food literacy; however, its use in progressing the measurement of food literacy is limited in a number of ways. Firstly, only two food literacy measurement tools were identified [16,17]. Secondly, the construct of food literacy is applied in many contexts beyond health, so to conflate its meaning with nutrition literacy presents a potentially reductionist, functional view of the role food plays in the lives of individuals, households, communities or nations, and therefore to the complexity of making food decisions. These are key tenets of food literacy, as distinct from nutrition literacy. While Yuen et al. provide a starting point for the appraisal of food literacy measures, their review is limited by a search strategy that dismissed existing tools that evaluate food literacy interventions. Although their review sought to identify all available publications detailing the measurement of food literacy, it is unlikely to have identified all adult tools. Furthermore, the field is rapidly evolving, and more food literacy measurement tools are likely to have been published since they reviewed the literature in January 2018. Additionally, previous reviews have not critiqued tools against an explicit definition and conceptualisation of food literacy. Doing so, makes it easier for practitioners, researchers and policy makers to make a judgement on the use and application of a tool in their practice. It also enables specific items within each tool to be critiqued.

Consequently, the aim of this paper is to review the literature to describe and analyse the measurement of adult food literacy. The objectives are to (1) identify tools that explicitly measure food literacy in adults; (2) summarise their psychometric properties (validity and reliability); and (3) critique tool items against the four domains and 11 components of food literacy, as conceptualised by Vidgen and Gallegos [2]. By doing so, the gap in existing food literacy measurement tools will be elucidated and assist in advancing the construct.

## 2. Materials and Methods 

This scoping review was planned and conducted using the Preferred Reporting Items for Systematic Reviews and Meta-Analyses (PRIMSA) guidelines [18].

### 2.1. Search Strategy

#### 2.1.1. Search Strategy and Information Sources

A systematic literature search was performed in seven databases (PubMed, Embase, ScienceDirect, Scopus, EBSCOhost, A+ Education, and ProQuest) up to 18 November 2018 to identify published tools that explicitly measure food literacy in adults. No limitations were placed on year of publication. The following search terms were used to conduct a full-text search in each database: “food literacy” AND “intervention*” OR “program*” OR “survey*” OR “tool*” OR “question*” OR “measur*” OR “scale*” OR “assess*”.

#### 2.1.2. Inclusion and Exclusion Criteria, and Their Application

Articles acquired through the search strategy were imported into EndNote and duplicates were removed. Inclusion and exclusion criteria were applied at two stages of the review (refer to Figure 1). In acknowledgement that the review was seeking both, papers that were explicitly measuring food literacy, and those that were including the measurement of food literacy as part of a range of measures, screening occurred at two stages, with greater specificity at each stage. At stage one, the titles and abstracts were screened and excluded if they (1) did not focus on an adult population, (2) were not in the English language, (3) were grey literature, or (4) did not report a quantitative measure; that is, if they did not (i) report on the development of a measurement tool; (ii) evaluate an intervention; or (iii) examine a food or nutrition related behaviour. At stage two, the full text of remaining papers was screened and excluded if (1) they met the exclusion criteria of stage one, (2) the full text was not available and (3) if they did not include a measure of food literacy. Studies were also excluded if they included a tool that had already been identified in the review. In such circumstances, the article that originally published the food literacy measurement tool was prioritised. Publications were included if the article (1) reported on a measurement tool that explicitly referred to ‘food literacy’ in its conceptualisation or development and (2) included access to all items within its food literacy measurement tool. Each article was screened against the inclusion and exclusion criteria by two reviewers (C.A., H.V.) independently at both stages. Discrepancies were discussed and resolved. 

### 2.2. Data extraction

Data was extracted from each article on: (1) characteristics of the study (including first author, country and purpose of the tool); (2) characteristics of the target group (including recruitment method, sample size, age, education level, ethnicity and socio-economic status); (3) type of food literacy outcome measure; and (4) psychometric properties (including underlying conceptual framework, content validity, face validity, construct validity and reliability). The items from each tool were also extracted to critique against the four domains and 11 components of food literacy. Tool items that were not listed in the article were back-referenced and sourced from the original publication or requested from the author.

### 2.3. Data Synthesis and Analysis

To ascertain content validity of the identified food literacy measures, three authors (C.A., H.V., D.F.) independently reviewed the questionnaire items against the Vidgen and Gallegos conceptualisation of food literacy [2]. Specifically, the items were coded against the four domains and 11 corresponding components of food literacy [2]. No assessment of risk of bias or study quality was undertaken due to the heterogeneity of study designs included in the review.

## 3. Results

A total of 12 studies describing 12 different tools met the inclusion criteria (Figure 1) [16,17,19,20,21,22,23,24,25,26,27,28]. Searches of the seven databases identified 360 unique records. After screening the titles and abstracts, 269 studies were excluded as they did not meet the inclusion criteria. The full text of the remaining 91 articles were assessed and 79 were excluded. Of those excluded, 20 were conference abstracts or did not have the full text available, two did not have the full text available in English, one reported on a food literacy measurement tool published elsewhere, one was in a school setting and 55 did not include a measurement tool that explicitly referred to food literacy in its conceptualisation or development. No additional studies were obtained through other sources.

### 3.1. Study and Tool Characteristics

Table 1 presents the tool characteristics extracted from each paper.

#### 3.1.1. Tool Purpose

Included papers had four distinct purposes, those reporting the development of a tool to explicitly measure food literacy or a part thereof (*n* = 5); food literacy strategy indicators (*n* = 1); tools developed to evaluate a food literacy intervention (*n* = 3); and tools to measure food literacy as a characteristic within a broader study (*n* = 3). Most tools (*n* = 7) were used to assess the food literacy of individuals or populations, including adults residing in a particular country [17,24,25,26], household food gatekeepers [28] and adults diagnosed with cancer [19]. One tool proposed key performance indicators for food literacy strategic directions in a state public health nutrition strategy [22]. Four were also used to compare food literacy levels between people of different socioeconomic status [19,24,25,28] and five examined the relationship between food literacy and food intake [19,26,28]. Six of the 12 identified tools were used to evaluate the effectiveness of food literacy interventions in Australia [20,21,27], Switzerland [16], Canada [22] and the United Kingdom [23].

#### 3.1.2. Sample Characteristics

The sample size of tool respondents ranged from 21 to 62,373 adults. The age of participants ranged from 15 to 96 years. In seven studies, at least 60% of respondents were female. Three studies [20,21,23] used their tool with a high proportion of participants from a low socio-economic background. Four studies were undertaken in Australia [20,21,27,28], five in Europe (France [25], Italy [17], the Netherlands [26], Switzerland [16], the United Kingdom [23]), one in Canada [22] and two were in the United States [19,24].

### 3.2. Alignment with the Four Domains and Eleven Components of Food Literacy

Table 2 summarises the alignment of each reviewed tool against the eleven components of food literacy, which can be collapsed into the four domains of planning and management, selection, preparation and eating. The coding of each item within these tools is detailed in supplementary material.

Six tools [17,21,24,26,27,28] captured all four domains of food literacy (i.e., planning and management, selection, preparation, and eating). Among these, the tools used by Begley et al. [21], Lahne et al. [24], Palumbo et al. [17], Poelman et al [26] and Wijayaratne et al. [28] were the most comprehensive, addressing seven or more of the 11 food literacy components. Component 3.1 was the most frequently included by the food literacy measures (*n* = 10) [17,20,21,22,23,24,25,26,27,28], while 3.2 [17,21] and 4.3 [24,26] were only captured in two tools. Overall, almost all tools included items to evaluate ‘preparation’ (*n* = 10) and most assessed ‘selection’ (*n* = 8) and ‘eating’ (*n* = 7). Fewer tools addressed ‘planning and management’ (*n* = 6). The Ontario Food and Nutrition Strategy food literacy indicators were dominated by process indicators such as the number of people accessing services or interventions which focused on food literacy [20]. 

#### 3.2.1. Planning and Management 

The ‘planning and management’ domain of food literacy encompasses the ability to prioritise time and money for food (1.1); plan food intake (formally and informally) so that food can be regularly accessed through some source, irrespective of changes in circumstances or environment (1.2); and make feasible food decisions which balance food needs (e.g., nutrition, taste, hunger) with available resources (e.g., time, money, skills, equipment) (1.3) [2]. Of the six tools that assessed this domain, most (*n* = 5) included items to assess 1.2 or 1.3. Component 1.2 was commonly assessed through the frequency of planning meals (*n* = 3) [21,24,28] and using a shopping list (*n* = 2) [21,28], including specifically to meet nutrition recommendations [21] in anticipation of distractions and then adjusting food decision accordingly [26]. One study assessed confidence planning meals [27]. Tools that measured 1.3, included items that measured confidence choosing foods that are the best value for money [17,21,28], maintaining a focus on healthy eating irrespective of cost [26] and deciding what to eat [17,24]. Measures also examined balancing time management regarding meal preparation with other responsibilities [24], and critiquing external influences, such as social marketing [17]. Three tools captured component 1.1. To do this, they gauged participants’ attitude towards prioritising time for cooking [24,28], or experience of running out of money for food [21]. 

#### 3.2.2. Selection

Measurement of the ‘selection’ domain of food literacy requires investigations into the ability to access food through multiple sources and know the advantages and disadvantages of these sources (2.1); determine what is in a food product, where it came from, how to store it and use it (2.2); and judge the quality of food (2.3) [2]. Seven of the eight tools that evaluated this domain assessed how well participants can source information about a food product (2.2). To do this, four tools [19,21,26,28] evaluated food label use, four [16,17,24,28] gauged food label reading confidence and one [19] included a label reading task to assess food label comprehension. Palumbo et al’s tool included a greater number of items to measure a wider range of food information topics (e.g., providence), and sources (e.g., digital media). Items used to measure this component tended to overlap with components 4.1 and 4.2, specifically the nutrition knowledge needed to interpret the label, and the motivation to select the healthier product. Three tools alluded to food safety as a component of ‘quality’ food [17,26], while one attributed ‘quality’ to ‘natural’ foods and those free of additives and preservatives [28]. The analysis revealed three measures captured component 2.1. Specifically, confidence shopping [27], where to source particular foods [24], and the social, economic and environmental impact of food choices [17].

#### 3.2.3. Preparation

‘Preparation’ was the most common domain captured in the identified food literacy measurement tools. The majority of tools (*n* = 10) assessed component 3.1 (i.e., make a good tasting meal from whatever food is available, including the ability to prepare commonly available foods, efficiently using common pieces of cooking equipment and having a sufficient repertoire of skills to adapt recipes, written or unwritten, to experiment with food and ingredients) via self-perceived confidence with cooking techniques (e.g., confidence using kitchen equipment) or meal preparation (e.g., confidence cooking from basic ingredients/following a simple recipe) [17,20,21,23,24,25,26,27,28]. Other items corresponding with 3.1 assessed confidence trying and preparing new foods [20,21], as well as attitude towards cooking (e.g., cooking enjoyment) [24,25,28]. Some took a particular focus on preparing healthy foods [21,23,26]. One tool included an inventory of key items of kitchen equipment [25]. Tools ranged considerably in their level of specificity from an overall statement about cooking in general [23] to over 30 items on particular ingredients and dishes [25], and their focus e.g., confidence, frequency, attitude or behaviour. Two tools measured component 3.2 (i.e., apply basic principles of safe food hygiene and handling). Begley et al. [21] asked participants how frequently they thaw meat at room temperature, while Palumbo et al. [17] gauged confidence accessing information about food safety and hygiene practices.

#### 3.2.4. Eating

Food literacy includes understanding that food has an impact on personal wellbeing (4.1), demonstrating self-awareness of the need to personally balance food intake, including knowing foods to include for good health, restrict for good health and appropriate portion size and frequency (4.2), and being able to join in and eat in a social way (4.3) [2]. Four tools measured component 4.1 by assessing how often participants consider healthy choices when eating or preparing a meal [17,21,27,28]. In this way, there was significant cross over with components in other domains. Palumbo included items related to specific individualised health needs, as opposed to population nutrition recommendations, which was more the focus of component 4.2 [17]. Six tools captured component 4.2, of these, two tools [16,19] included knowledge questions relating to portion sizes and national dietary guidelines. One tool was tailored to the context in which it was administered and included specific nutrition knowledge questions related to the target audience of the food literacy intervention [27]. The remaining tools assessed self-perceived confidence with nutrition knowledge [17,28] or competence balancing food intake [16,26]. Two food literacy measurement tools investigated social eating and relied on self-reported attitude toward shared eating occasions [24,26]

### 3.3. Psychometric Properties of Process Evaluation Measures

Most articles (*n* = 8) reported some psychometric properties of their food literacy measurement tool (see Table 1 and Table 3). Nine tools were underpinned by a conceptual framework [16,17,21,22,24,25,26,27,28]. Of these, seven referred to the empirical definition of food literacy by Vidgen and Gallegos [17,21,24,25,26,27,28], one the definition by Krause [31] and one used Desjardin’s definition [44]. Health literacy definitions were also used, specifically Nutbeam’s [51,52], by Krause [16], Sørensen [53], Poelman [26] and Palumbo [17]. Lahne [24] used a conceptualisation of food agency [32,33] which included food literacy. Tools developed for the purpose of evaluating a food literacy intervention were the least likely to be underpinned by a conceptual framework.

Content validity was assessed in nine articles through dietitians, public health experts and food literacy experts [17,20,21,22,26,27], and by pooling items from pre-existing tools [16,21,24,26,28] or population monitoring and surveillance systems [22] (see Table 1 and Table 3). Table 3 reports the face, content and construct validity, and reliability of reviewed tools. Eight tools [16,17,20,21,24,26,27,28] reported reliability of their measure using Cronbach’s alpha, with internal consistency ranging from α = 0.76–0.95. The highest reported rates were α = 0.94 for the confidence in cooking, shopping, planning and purchasing scale in the Wallace et al. tool [27], and α = 0.912 for the general food literacy scale in the Palumbo et al. tool [17]. The weakest internal consistency was reported for the selection, plan and manage, and preparation scales, at α = 0.76, 0.79 and 0.81 respectively, in the Begley et al. tool [21]. Eight tools were face validated [16,17,20,21,24,25,26,28]. Five tools reported examining construct validity [16,17,21,24,26]. In order to test for construct validity, two tools regressed against gender and education, assuming food literacy would be higher among females and those with a higher education [16,24]. Overall, five food literacy measurement tools reported content, face and construct validity, as well as reliability [16,17,21,24,26].

### 3.4. Outcome Measure

Of the 12 tools identified, five evaluated the impact of food literacy on dietary intake (see Table 1). Dietary outcomes were assessed in various ways, including specific foods (e.g., core food groups, fruits, vegetables, fish, herbs/spices/salt, unsaturated spreads and oils, sugar-sweetened beverages), food types (e.g., fibre, discretionary choices, snacks), or nutrients (energy, macronutrients and micronutrients) [19,20,23,26,27]. Only two tools cited a validated dietary intake measure, including the US Department of Agriculture’s five-step multiple-pass method [20] and a validated food frequency questionnaire [26].

## 4. Discussion

This scoping review identified 12 tools [16,17,19,20,21,22,23,24,25,26,27,28] measuring food literacy in adults. Each tool was appraised against the Vidgen and Gallegos [2] conceptualisation of food literacy and evaluated according to psychometric properties. 

This review revealed that this is a rapidly emerging area of public health nutrition activity, with all of the papers being published in the last three years. Of particular note is that all papers reporting the development of a tool to explicitly measure food literacy, or a part thereof, had been published in the last two years. These papers reported the most rigorous processes of development.

Food literacy measurement tools are becoming increasingly multidimensional, which reflects emerging theories that define the construct with multiple domains [10,11,54,55,56]. A 2012 review of 21 food literacy interventions targeting disadvantaged youth found that 90% of evaluation tools measured the ‘preparation’ domain, while only 30% measured ‘planning and management’ [57]. Although the present review found that ‘preparation’ still dominates most tools (*n* = 11; 85%) and ‘planning and management’ remains the least captured (*n* = 7; 54%), there is now a more even distribution of domains within and among tools. Half of the reviewed tools [17,21,24,26,27,28] captured all four domains of food literacy. These findings suggest food literacy measurement is extending beyond the cooking and meal preparation paradigm and recognising recent theoretical advances.

Existing food literacy measurement tools generally allude to a conceptual framework; however, interpretation and application of the theory is still limited. While the majority of tools (7/12) used the Vidgen and Gallegos empirical definition of food literacy [2], indicating greater agreement on its conceptualisation, there was significant variation in how the definition was applied. Alignment of the tools to the four domains of food literacy was typically clear; yet, there was difficulty coding the tool items against the 11 components. This was especially demonstrated by the disagreement between the coding of items by the tool developers and this paper’s review team, which included the developer of the cited definition. This could be attributed to the highly inter-related nature of the food literacy components [2] and construct as a whole [10], or may point to a need for the components to be more explicit and independent of each other. As such, it may be necessary for a food literacy measurement tool to address all 11 components of food literacy in order to appropriately capture the construct, rather than conflate this to its four domains. No tool was found to align with all 11 components, and the five [17,21,24,26,28] most comprehensive tools missed at least three. This may also reflect the conceptual nature of the Vidgen and Gallegos components and their lack of testing quantitatively. Reviewed papers that described the development of a tool to explicitly measure food literacy or a component of it [16,17,21,24,26] also described the process of beginning with a larger pool of items, which were later discarded as they moved through various stages of validation. This review included only the final set of items. It may be that items aligning with missing components were in their original item pool. 

Not only did tools vary considerably regarding the extent to which they included domains and components of food literacy, but items measuring specific components also varied between tools. That is, questions used to measure what is indicative of food preparation, for example, were very inconsistent. This variation included the focus of questions. More tools used subjective (self-report) measurement approaches, as opposed to objective (task-based) items, when measuring food literacy. According to the literature, self-reported confidence in food preparation, cooking and label reading does not necessarily translate to everyday use of such skills [12,58,59,60]. Similarly, self-reported food safety practices are particularly prone to social desirability bias and inaccurate responses [12]. Nevertheless, most food literacy measurement tools required participants to self-report their confidence undertaking such behaviours. Only one tool [19] used a task-based item to assess label reading skills (component 2.2) and thereby increased reliability of results. Moreover, many food literacy measurement tools included subjective attitudinal items pertaining to food mavenism (e.g., “I find cooking a very fulfilling activity” [24]; “I consider myself to be an excellent cook” [28]) to gauge the ‘preparation’ domain. While food mavenism and pleasure may be a positive predictor of food knowledge and involvement [61,62], a food literate person does not have to be a ‘food maven’. In fact, both food literacy experts and young people experiencing disadvantage agreed food preparation skills only needed to be “basic” to support needs [63]. Greater consideration of objective items is required to reduce social desirability bias and improve the validity of food literacy measurement tools. Items were generated using existing tools or expert consensus rather than empirical observation of people’s lived experience or evidence of behaviours that result in an improved dietary outcome.

Although most food literacy measurement tools have been assessed for validity and reliability, they are limited by inadequate validation methods and narrow sample demographics. In order to test for construct validity, some tools regressed food literacy against social determinants of health, such as education [16,24] and income [24]. While poverty, social exclusion, social support, geography and transport can influence the development of food literacy, people from all socio-economic backgrounds are capable of demonstrating food literacy [63]. Therefore, validating the food literacy construct against education and income may not be accurate. Additionally, existing food literacy measurement tools were content validated with dietitians and public health experts [20], food literacy experts [16,17,21,26,27] and pre-existing scales [16,19,20,21,24,28]. No tools were content validated with the general population; however, such validation may be important as food literacy is an everyday practice. Furthermore, sample respondents of the food literacy measurement tools were relatively homogenous. Indeed, most tools were tested with highly educated females living in Western countries, and none were applied across multiple contexts. Because of these limitations, it is difficult to determine the true validity and applicability of existing food literacy measurement tools in different contexts, in particular, if it is possible to compare food literacy between groups e.g., different countries.

All reviewed measures were developed or used within the health paradigm. As such, they reflect food literacy for dietary outcomes, rather than a broader conceptualisation of food wellbeing. The limited food literacy outcomes measured by existing tools further constrain use in diverse contexts. A comprehensive tool should encompass multiple indicators to measure against food literacy, including diet quality [63], intake of ultra-processed foods [64], food security [65] and sustainable eating [66]. Nevertheless, current tools are merely measured against food intake outcomes. In order to maximise relevance and applicability, food literacy measurement tools should measure against a range of outcomes. Our review chose to include only those papers that had explicitly mentioned food literacy. Another approach could have been to look for existing measures of each individual component. This may have resulted in a broader set of outcomes, e.g., food waste, being identified.

Although it is challenging to conclude which existing food literacy measurement tool is best, the findings of this review underscore the importance of using a multidimensional tool validated in the appropriate context. When choosing a suitable tool, those that capture the greatest number of food literacy components should be prioritised. More comprehensive food literacy measurement tools are more likely to accurately capture the interrelated nature of the construct. In this review, the tools used by Begley et al. [21], Lahne et al. [24], Palumbo et al. [17], Poelman [26] and Wijayaratne et al. [28] were found to be the most comprehensive. To further maximise validity, tool selection should be contextually driven. Existing tools have been validated with the general adult population [17,24,25,26], participants from low socio-economic backgrounds [20,21,23], household food gatekeepers [28] and adults diagnosed with cancer [19]. Furthermore, they have been applied in Western countries, such as Australia [20,21,27,28], France [25], Italy [17], the Netherlands [26], Switzerland [16], the United Kingdom [23] and the United States [19,24]. Nevertheless, there are many contexts in which food literacy measurement tools have yet to be validated and further research is required to address these gaps. Indeed, application of highly comprehensive and contextually-relevant tools will enhance the validity of food literacy measurement and assist in advancing the construct. Only one tool considered performance indicators of food literacy at a population level within existing monitoring and surveillance systems [22]. Given that food literacy is conceptualised as existing at “individual, household, community or national level” [2], and is included in an increasing number of key national and state public health nutrition plans [67], this is an aspect of measurement which requires greater attention.

In considering these findings, certain limitations of this review should be noted. Only peer-reviewed journal articles published in English were included, which may have introduced selection bias. Furthermore, no reporting guidelines were used to evaluate the quality of studies. Finally, the food literacy measurement tools were coded from a nutrition paradigm, which may have incurred biases. Despite these limitations, this review is strengthened by its compliance to the PRISMA guidelines and robust search strategy conducted in seven databases. Moreover, this review appraised existing food literacy measurement tools against the Vidgen and Gallegos food literacy conceptualisation and offers valuable insight into the current state of food literacy measurement. It should also be noted that this review examined only those tools targeting adults. During the screening process, many additional tools targeting children and school settings were identified, but excluded (refer to Figure 1). Analysis of these tools would further add to the field, particularly within the context of education. 

## 5. Conclusions

There are currently 12 tools available that measure food literacy among adults. Tools varied considerably in their item type, indicating there is still time before we are able to compare food literacy interventions, determine their effectiveness, report on populations over time, and most importantly, determine the relationship between food literacy and food intake. While most tools capture all four domains of food literacy, the application of theoretical frameworks is limited. To date, no tools have been explicitly built from the Vidgen and Gallegos conceptualisation or crafted to capture all 11 components of food literacy. Furthermore, existing tools have been validated across limited contexts and have relied heavily on self-report methods that are prone to bias. Existing tools must be applied with care and further research is required to develop comprehensive tools that are contextually valid. This review helps advance the measurement of food literacy and provides useful information that will assist researchers in selecting and developing validated food literacy measurement tools.

## Figures and Tables

**Figure 1 nutrients-11-00801-f001:**
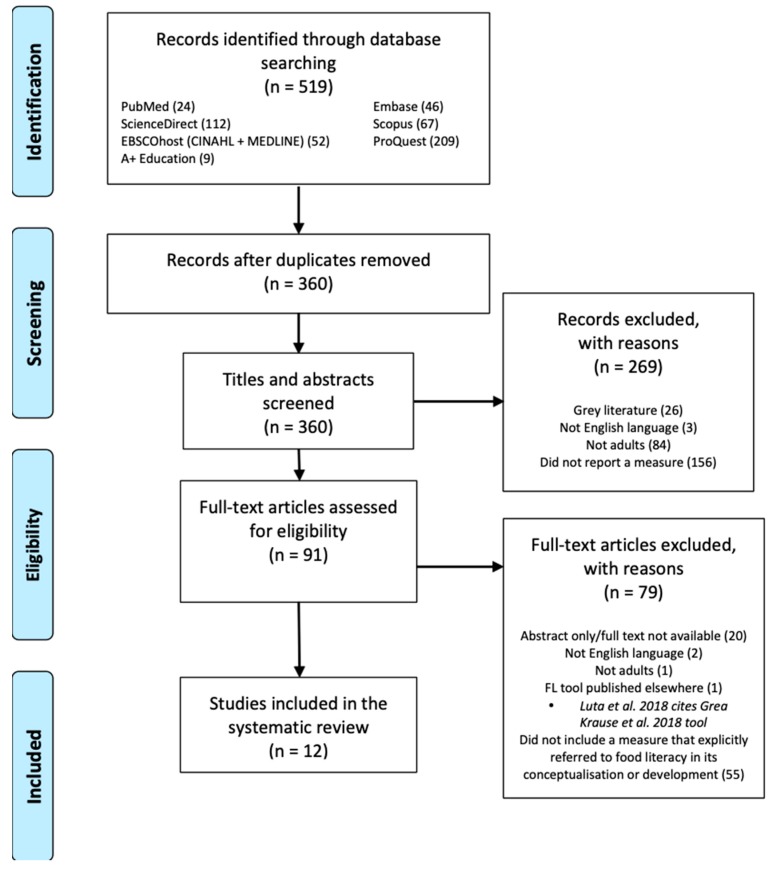
PRISMA flow chart.

**Table 1 nutrients-11-00801-t001:** A summary of key characteristics and underlying conceptual framework of reviewed food literacy measures.

Reference	Purpose	Name of Tool and Number of Items	Definition or Conceptual Framework of Food Literacy	Item Generation	Sample Characteristics	Sampling Method	Administration	Outcome Measure
***Papers reporting the development of a tool to explicitly measure food literacy or a part thereof***								
**Begley (2018) Australia [21]**	To develop and validate a self-administered questionnaire to measure food literacy behaviour for a food literacy program evaluation.	Food Literacy Behaviours tool 14 items	Vidgen & Gallegos (2014) [2]	Townsend, et al (2003) [29] Phelps et al (2017) [30]	Participants of Food Sensations® for Adults program ≥18 years old *n* = 1012 82% female 71% from low- or middle-income status postcodes Most represented age: 26–35 years 28.5% unemployed 25.2% university degree 55.2% born in Australia 7.1% Aboriginal or Torres Strait Islanders	All program participants	Self-administered questionnaire	
**Krause (2018) Switzerland [16]**	To evaluate the measurement properties of the Short Food Literacy Questionnaire (SFLQ).	Short Food Literacy Questionnaire 18 items	Krause, Sommerhalder, & Beer-Borst (2016) [31]	A search of scientific publications in German, English and French: search terms: health literacy, nutrition literacy, food literacy, instrument, questionnaire, survey, valid *, reliab * Reported in Krause, Sommerhalder, & Beer-Borst [31]	Participants of a workplace intervention trial to reduce salt intake in Switzerland. *n* = 350 16–65 years 62% female 77% tertiary educated 2015–2016	All study participants	Self-administered questionnaire in electronic or paper format	Health literacy Nutrition knowledge Salt intake
**Lahne (2017) USA [24]**	To employ the Food Agency paradigm to develop scale items assessing the individual’s perceptions of their cooking skill and ability to prepare food. Food literacy was explicitly described as being an element of food agency.	Cooking and Food Provisioning Action Scale (CAFPAS) 28 items	Food Agency paradigm (Trubek, Carabello, Morgan, & Lahne, 2017; Wolfson et al., 2017) [32,33] Vidgen (2014) definition of food literacy [2]	Initial item pool developed based on related qualitative research (Carabello, 2015 [34]) and published research on food and cooking behaviour (Bell and Marshall, 2003, Bisogni et al., 2005, Bisogni et al., 2007, Hartmann et al., 2013, Jastran et al., 2009, Marshall and Bell, 2004, Sobal and Bisogni, 2009, Sobal et al., 2014). [35,36,37,38,39,40,41,42]	US Development sample: *n* = 445 mean age 34.8 Validation sample: *n* = 498 Mean age = 35.4 57% female 71% White 72% college educated 52% annual income under US$50,000	Development: convenience sampling from Universities Validation sample: recruited using Amazon.com’s Mechanical Turk (MTurk) “Human Intelligence Task” system	Self-administered questionnaire online	Frequency of home cooking
**Palumbo (2017) Italy [17]**	To develop a self-reported food literacy assessment tool which measures an individual’s level of food literacy and investigates the main consequences of inadequate food literacy.	The Italian Food Literacy Survey 47 items	Vidgen & Gallegos (2014) [2]	Expert consensus	Pilot convenience sample were clients of dietitians involved in the project. Italy *n* = 158 50% female Mean age = 43 22.8% tertiary educated	Convenience sample	Self-reported dietitian administered survey	Newest Vital Sign (Health Literacy) [43] BMI
**Poelman (2018) Netherlands** [26]	To develop and validate the Self Perceived Food Literacy Scale to assess individual food literacy.	Self-perceived food literacy (SPFL) scale 29 items	Vidgen & Gallegos (2014) [2]	Expert consensus	Study 1: Dutch adults *n* = 755 90.7% female Mean age: 44.8 59.1% highly educated 97.4% Dutch Study 2: Dutch dietitians *n* = 207 98.5% femaleMean age: 43.4 100% highly educated 95.6% Dutch	Study 1: recruited via the Facebook page and Twitter account of The Netherlands Nutrition Centre. Study 2: recruited via the monthly Netherlands Association of Dietitians newsletter.	Self- administered online survey	Dietary intake of fruit, vegetables, fish, sugar sweetened beverages, ‘large’ unhealthy snacks (e.g., pizza slice, piece of pie) and ‘small’ unhealthy snacks (e.g., biscuit, candy)
***Food literacy strategy indicators***								
**Boucher (2017), Canada [22]**	To describe the Ontario Food and Nutrition Strategy (OFNS), which integrates multiple sectors and determinants, available indicators through existing information systems and activity across Canada to develop a strategy and surveillance system. Three key strategic directions were identified, one of which is “food literacy and skills”.	OFNS Food Literacy Indicators. 4 items	Desjardins E et al. Locally Driven Collaborative Project. 2013 [44].	Five step process undertaken by the OFNS Advisory Group, which began with an environmental scan of existing system level data, development of assessment criteria, face validity, and finally feasibility.				
***Tools developed to evaluate a food literacy intervention***								
**Barbour (2016) Australia [20]**	To assess the impact of a food literacy program on (i) dietary intake (ii) diet quality, (iii) cooking confidence and (iv) food independence		No definition	4 food literacy questions sourced from a questionnaire used previously with adults attending a cooking skills intervention [45]	2014 Participants of food literacy programme: FoodMate by SecondBite in Australia *n* = 21 (aged 16–25 years); 8 completed cooking confidence questions median age = 20 47% experiencing homelessness 45% experiencing food insecurity	All programme participants	Self-administered or case worker administered face-to-face questionnaire	Consumption of core food groups, unsaturated spreads and oils, discretionary choices, sugars-sweetened beverages, energy, carbohydrate, protein, total fat, saturated fat, fibre, sodium, calcium, iron, folate and vitamin C Food independence
Hutchinson (2016) UK [23]	To evaluate the impact of Jamie Oliver’s Ministry of food Cooking Course, which aims to lead to improved dietary intake and improved food literacy.	Ministry of Food cooking programme evaluation Single question on cooking confidence	No		Participants in the Jamie Oliver Ministry of Food cooking programme in the UK. *n* = 795 ≥16 years old (81% 20–64 years) 57% female 85% white 24% from deprived area 63% no disability 2010–2014	All programme participants	Self-administered questionnaire	Portions of fruit and vegetables Frequency of snacks
Wallace (2016) Australia [27]	To evaluate the effectiveness of a 4-week nutrition education intervention to determine long term efficacy of food literacy intervention on long term food literacy.	11 items quantitative questionnaire and qualitative focus groups.	Vidgen and Gallegos (2014) [2]		Participants in a 4 week dementia and nutrition education intervention. Participants were healthy, independently living individuals without a dementia diagnosis but an interest in the subject *n* = 72 81% aged >61 70% female 34% with CVD	All participants	Self-reported questionnaire At baseline (*n* = 72), post evaluation (n = 66) and >3 months post evaluation (*n* = 42)	Fruit and vegetable variety and intake, herb, spice and salt use, and trimming fat behaviours
***Tools to measure food literacy as a characteristic within a broader study***								
**Amuta-Jimenez (2018) USA [19]**	To examine: differences between food label use and food label literacy between participants who had a cancer diagnosis and those without Sociodemographic correlates and health related correlates of food label use and literacy Potential associations between food label use/literacy and food choice	1 question on food label use and 6 food label literacy questions within the Health Information National Trends Survey (HINTS) [46]	Food label literacy conceptualised as a subset of health literacy (no citation given)	Food label literacy questions from Newest Vital Signs [43]	2013-14 HINTS participants in US ≥18 years old *n* = 260 adults 61% female 54.6% college educated 76.2% White 52.1% annual income under US$50,000		Mail survey	Vegetables and fruit Sugary drinks
Mejean (2017) France [25]	To assess which socio-economic indicators are independently associated with various dimensions of food preparation.	Food preparation behaviours Approx. 49 items	Various conceptualisations food preparation behaviours including “preparation” domain of food literacy Vidgen & Gallegos (2014) [2]	Conceptualisations informed the development of questions	Participant data sourced from the NutriNet-Santé cohort study. France *n* = 62,373 78% female 2009	Multimedia campaign of adults over 18 years	Self-administered online questionnaire	
Wijayaratne (2018) Australia [28]	To examine how dietary gatekeeper’s intentions to prepare a healthy diet and the subsequent satisfaction that a healthy diet is achieved, is influenced by their food literacy and by barriers to healthy eating.	Gatekeeper food literacy questionnaire. 29 items	Vidgen and Gallegos (2014) [2]	Existing scales, specifically: the food-related lifestyle framework and research of Grunert et al. (1993), Brunsø et al. (2004) and Buckley et al. (2007) and household gatekeeper food acquisition and transformation work of Reid et al. (2015). [47,48,49,50]	Dietary gatekeepers *n* = 756 31% aged 41-50 years 70.9% female 31.9% university degree 43.5% household income ≤$60,000 22.6% with a medical condition that affects household eating	Recruited via email through the Global Market Insite.	Self-administered online survey	Diet satisfaction; Attitude to healthy eating; Perceived behavioural control; Intention to prepare a healthy diet;

**Table 2 nutrients-11-00801-t002:** The alignment of questions within each reviewed food literacy measurement tool against the four domains and 11 components of food literacy as conceptualised by Vidgen and Gallegos (2014) (refer to Appendix A for coding of specific questions).

First Author (Year), Country	Domains and Components of Food Literacy											TOTAL Components (Domains)
	1. Planning and management			>2. Selection			>3. Preparation		>4. Eating			
	>1.1	>1.2	>1.3	>2.1	>2.2	>2.3	>3.1	>3.2	>4.1	>4.2	>4.3	
***Papers reporting the development of a tool to explicitly measure food literacy or a part thereof***												
**Begley (2018)** **Australia [21]**	✓	✓	✓		✓		✓	✓	✓			7 (4)
**Krause (2018) Switzerland [16]**					✓					✓		2 (2)
**Lahne (2017) USA [24]**	✓	✓	✓	✓	✓		✓				✓	7 (4)
**Palumbo (2017) Italy [17]**			✓	✓	✓	✓	✓	✓	✓	✓		8 (4)
**Poelman (2018) Netherlands [26]**		✓	✓		✓	✓	✓			✓	✓	7 (4)
***Food literacy strategy indicators***												
**Boucher (2017) Canada [22]**							✓					1 (1)
***Tools developed to evaluate a food literacy intervention***												
**Barbour (2016) Australia [20]**							✓					1 (1)
**Hutchinson (2016) UK [23]**							✓					1 (1)
**Wallace (2016) Australia [27]**		✓		✓			✓		✓	✓		5 (4)
***Tools to measure food literacy as a characteristic within broader study***												
**Amuta-Jimenez (2018) USA [19]**					✓					✓		2 (2)
**Mejean (2017) France [25]**							✓					1 (1)
**Wijayaratne (2018) Australia [28]**	✓	✓	✓		✓	✓	✓		✓	✓		8 (4)
**TOTAL** **measures**	3	5	6	3	7	3	10	2	4	6	2	

**Table 3 nutrients-11-00801-t003:** Content, face and construct validity and reliability of reviewed food literacy measures.

First author (Year), Country	Content Validity: Inclusion of All Intended Domains? *	Face Validity	Construct Validity: Compared to Other FL Indices?	Reliability
***Papers reporting the development of a tool to explicitly measure food literacy or a part thereof***				
Begley (2018) Australia [21]	Tool based off EFNEP behaviour checklist and validated with four food literacy experts.	Yes	Yes; pre- and post-program food literacy behaviours.	Cronbach’s alpha: Plan & Manage (0.79) Selection (0.76) Preparation (0.81)
Krause (2018) Switzerland [16]	Tool was constructed with items adapted from different existing instruments on health and nutrition literacy, as well as newly developed items. Tool addresses functional, interactive and critical FL.	Yes	Yes; examined tool’s association with gender, health literacy, education, and nutrition knowledge.	Cronbach’s alpha: All 12 items (0.82)
Lahne (2017) USA [24]	Tool was constructed from existing qualitative research (Carabello, 2015) and published research on food and cooking behaviour. Tool aligns with the four potential dimensions of food agency.	Yes	Yes; food involvement, self-reported meals cooked at home, age, income, sex, race and education.	Scale exceeds α > 0.70
Palumbo (2017) Italy [17]	The domains of ‘preparation’ and ‘eating’ were aggregated in the tool. Survey items were generated by experts in food literacy and health literacy.	Yes	Yes; compared with the Newest Vital Sign screening tool (Weiss et al., 2005), as well as gender, age, education, social status and financial deprivation.	Cronbach’s alpha: General FL (0.912) Plan and Manage FL (0.879) Select and Choice FL (0.881) Prepare and Consume FL (0.893) NVS (0.870)
Poelman (2018) Netherlands [26]	Tool items were generated by experts in food literacy and using existing literature.	Yes	Yes; validated the scale against psychological constructs which are well known for their positive (self-control) and negative (impulsiveness) correlation with healthy food consumption (convergent and divergent validity).	Cronbach’s alpha: Overall scale (0.83) Food preparation skills (6 items, α: 0.78) Resilience and resistance (6 items, α = 0.80) Healthy snack styles (4 items, α = 0.58) Social and conscious eating (3 items, α = 0.69) Social and conscious eating (3 items, α = 0.69) Examining food labels (2 items, α = 0.90) Daily food planning (2 items, α = 0.72) Healthy budgeting (2 items, α = 0.85) Healthy food stockpiling (4 items, α = 0.81)
***Food literacy strategy indicators***				
Boucher (2017) Canada [22]	Examination of existing tools within existing population monitoring and surveillance systems.	Yes	Not reported	Not reported
***Tools developed to evaluate a food literacy intervention***				
Barbour (2016) Australia [20]	Original tool validated for cooking skills with dietitians and public health experts.	Yes	Not reported	Original tool Cronbach’s alpha: Confidence (0.86) Knowledge (0.84) Spearman correlation coefficients were in the range 0.46–0.91 and were statistically significant (*p* < 0.001)
Hutchinson (2016) UK [23]	Not reported	Not reported	Not reported	Not reported
Wallace (2016) Australia [27]	Content validity was determined by an expert in nutritional aspects of vascular disease and dementia research.	Not reported	Not reported	Cronbach’s alpha: Attitudes to healthy eating and cooking (0.85) Confidence in cooking, shopping, planning and purchasing (0.94)
***Tools to measure food literacy as a characteristic within broader study***				
Amuta-Jimenez (2018) USA [19]	Not reported	Not reported	Not reported	Not reported
Mejean (2017) France [25]	Not reported	Yes	Not reported	Not reported
Wijayaratne (2018) Australia [28]	Based on existing scales with known psychometric properties.	Yes	Not reported	Composite reliabilities Cooking and nutrition capability (0.932) Informed food choices (0.919) Making time (0.844) (not) convenience foods and cooking (0.831) Fresh food focus (0.878) Planned meals (0.850)

* refer to item generation in Table 1.

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
