# Peer review of "A Scoping Review of the Validity, Reliability and Conceptual Alignment of Food Literacy Measures for Adults"

_nutrients, 2019, doi:10.3390/nu11040801_

Round 1
Reviewer 1 Report
The paper entitled “A Systematic Review of the validity, reliability and conceptual alignment of food literacy measures for adults” is a very interesting paper analyzing and describing the measurement of adult food literacy. This is a well-written work although there are some concerns about several issues that the authors should carefully tackle before considering the manuscript for publication.
1) Authors conducted a systematic review on a very novel issue in nutritional epidemiological research, that is, the measurement of adult food literacy. Although it is an important aspect of this field and they used an appropriate research methodology, the type of review is not likely the most adequate approach for this particular topic. Systematic review should be used to appraise and synthesize research evidence in order to identify reliable and quality data on a particular issue. In this sense, a considerable accumulative evidence is required. It explains that the authors were not be able to assess study quality and risk of bias.
2) Authors should consider that their review is closer to a Scoping review than a Systematic review. This type of review is aimed at assessing the potential size and scope of the available literature on a particular issue. The main objective of a scoping review is to identify the nature and extent of resources available, including ongoing research. It reveals that the gaps of knowledge for further research in order to have a better understanding of this topic.
Author Response
Thank you for your comments on our paper. We agree that the paper is more accurately described as a scoping review. To reflect this we have changed the title to:
A Scoping Review of the validity, reliability and conceptual alignment of food literacy measures for adults.
and removed the term "systematic" in lines 11,49 and 62.
In lines 69 and 264 we have replaced the term "systematic" with "scoping" in lines 69 and 264.
These changes have been highlighted in the revised manuscript as track changes
Reviewer 2 Report
Authors did a nice job of providing the theoretical background on food literacy and evaluating current measurement tools based on how well they align with theory and how well they perform in practice. Discussion and conclusions also did a good job at outlining current weaknesses of food literacy measurement tools and describing where the literature needs to go next.
Author Response
Thank you for your comments on our paper
Reviewer 3 Report
Very interesting paper and I agree with the authors there is certainly a need for a literacy measure. However, I question why the authors chose food literacy, rather than nutrition literacy. As the authors rightly stated, the increase in ill health and confusion with regards to food information has created the need to assess the impact of food on well being. However, I would have leaned more towards nutrition literacy which emphasises one aspect of the food literacy concept - selection. It is a little late now to change the focus on the study, and neither am I asking this. However, it would be good if at the start of the paper, the authors would acknowledge this concern by better justifying their choice to construct in lieu or other constructs such as nutrition literacy. This is my major issue with the paper.
Author Response
The following text has been added to line 51 of the introduction:
A recent review by Yuen, Thomson & Gardiner 15 critically appraised the psychometric properties of 13 existing measures of adult nutrition literacy and food literacy; however, it’s use in progressing the measurement of food literacy is limited in a number of ways. Firstly, only two food literacy measurement tools were identified 16-17. Secondly, the construct of food literacy is applied in many contexts beyond health so to conflate its meaning with nutrition literacy presents a potentially reductionist, functional view of the role food plays in the lives of individuals, households, communities or nations, and therefore the complexity of making food decisions. These are key tenets of food literacy as distinct from nutrition literacy. While Yuen et al. provide a starting point for the appraisal of food literacy measures, their review is limited by a search